# Qualitative study of the consequences of vaccine-induced immune thrombocytopenia and thrombosis: the experiences of family members

Paul Bennett ![ORCID],[1] Filiz Celik,[2] Jenna Winstanley,[3] Beverley J Hunt,[4] Sue Pavord[5]

[1]Psychology, Swansea University, Swansea, UK
[2]Swansea University, Swansea, UK
[3]Private practice, London, UK
[4]Thrombosis and Haemophilia Centre, Kings Healthcare Partners, London, UK
[5]Oxford University Hospitals NHS Foundation Trust, Oxford, UK

**Correspondence to**
Professor Paul Bennett;
p.d.bennett@swansea.ac.uk

## ABSTRACT

**Objectives** To explore the experiences of family members of patients who died or survived following a diagnosis of vaccine-induced immune thrombocytopenia and thrombosis (VITT).

**Design** A semistructured qualitative study, conducted via Zoom.

**Setting** Participants discussed their experiences during hospitalisation and following discharge.

**Participants** Sixteen family members of patients with VITT (survivors=11; bereaved=5), recruited via a Facebook support group and advertising on Twitter.

**Results** Analysis identified two themes common to both groups of participants: the stress of hospitalisation and the experience of multiple losses. A third theme, living with VITT, was unique to the survivor group and a fourth, battling against the system, was predominantly reported by bereaved participants.

**Conclusions** This is a significantly challenged group of people, with multiple emotional, financial, social and psychological losses. These losses have been compounded by experiences of limited governmental and societal recognition of the problems they face.

## STRENGTHS AND LIMITATIONS OF THIS STUDY

⇒ The first study to explore the impact on the families of patients surviving and dying of vaccine-induced immune thrombocytopenia and thrombosis (VITT).
⇒ The sample included approximately 10% of the surviving relatives of all UK patients who died from VITT.
⇒ Although attempts were made to widen recruitment, the majority were from a VITT support group provided by Thrombosis UK and in response to tweets via Twitter (X).

## METHOD

Qualitative interviews were conducted 15–24 months following a diagnosis of VITT.

### Patient and public involvement

The study protocol was discussed separately with several relatives of patients with VITT prior to conducting the research. In addition, all participants had repeated sight of the paper as it developed to ensure their agreement with the themes identified, quotations cited and degree of anonymity provided.

### Recruitment and sample

Participants were family members of patients with a confirmed diagnosis of VITT, recruited via social media: (1) a closed VITT Facebook support group hosted by Thrombosis UK, (2) the Thrombosis UK Twitter feed and (3) re-tweets by individuals known to members of the Facebook group or who read the Twitter feed. The final sample comprised 16 participants. Eleven were family members of surviving patients with VITT (nine partners, two adult daughters). Five were bereaved. There were 13 women and 3 men, with a median age of 53 years (range 21–59).

### Method

Interviews were conducted between August 2022 and July 2023. People who contacted the

## INTRODUCTION

We have previously explored the experiences of patients diagnosed with vaccine-induced immune thrombocytopenia and thrombosis (VITT) over the 18 months since their acute presentation.[1] During these interviews, it became apparent that the experiences of patients' families (including health, financial and social concerns) largely mirrored those of the patients, although this was not formally explored in the study. Furthermore, a mortality rate of more than 20%[2] necessarily meant that some families had to cope with the sudden and unexpected death of a family member, often at a young age. The present study aimed to explore the experiences of both bereaved and survivor families.

study team were sent a Participant Information Sheet and confirmed their willingness to participate by making further contact with the study lead. After providing informed consent, participants took part in a semistructured virtual interview on Zoom, addressing their experiences of hospitalisation and the longer-term impact of living with a patient with VITT or of their bereavement (see online supplemental appendix 1). Three people (PB, FC, JW) trained in clinical/counselling psychology and with previous experience of VITT,[1] conducted the interviews.

## Data analysis

Interviews were recorded on Zoom and a 'clean' transcript determined, after which the original recordings were deleted. All identifying elements were removed from the transcripts, which were analysed by PB using inductive thematic analysis.[3] Audio recordings were listened to repeatedly to identify relevant initial codes and textual units. Extracts and phrases were used to identify potential themes, with relevant quotes gathered within identified themes. Themes and subthemes derived from the data were checked and validated by FC; any differences were resolved through discussion.

## RESULTS

Of the themes identified, theme 1, the stress of hospitalisation, has commonalities across both groups of participants. Theme 2, living with VITT, is unique to families of survivors. Theme 3, multiple losses, has commonalities across both groups. Theme 4, battling against the system, is predominantly reported by bereaved participants. Table 1 provides an overview of the themes and subthemes.

### Theme 1: the stress of hospitalisation
#### The dawning light of seriousness

Most participants were concerned at the initial onset of symptoms. However, this anxiety increased significantly when participants became aware of the severity of their family member's diagnosis, often given over the phone due to COVID-19 pandemic imposed visiting limitations.

| Table 1 | Key themes and subthemes |
|---|---|
| **Theme** | **Subtheme** |
| The stress of hospitalisation | The dawning light of seriousness |
| | Breaking the bad news |
| | Waiting for news |
| Living with VITT | Coming home is not the end |
| | The whole family has to adjust |
| Multiple losses and some gains | Loss of a loved one |
| | Loss of a future |
| | From trauma, growth |
| Battling against the system | |
| VITT, vaccine-induced immune thrombocytopenia and thrombosis. | |

Anxiety was frequently exacerbated by incomplete understandings of the medical condition:

> I had a consultant phone, and she was saying, … She's reading through a big, long list of things. … She's going down this list. And then she says, 'Oh, and she's had a stroke.' That's the one word that my mind picked up on because that's what I knew about. … She went, 'Yes, but don't worry about that. That's the minor thing what's wrong with her.' That's when you think, God … The stroke is the minor thing! … what's the rest of it!? (5: survivor)

Many relatives of survivors were fearful they would die. As one participant noted, 'I suddenly woke up, like bang at four o'clock in the morning, and I thought… she could actually be dead now. She could be dead now.' (7: survivor). For those who did die, there was little or no time for preparation. Four of the five participants were rung at home shortly after admission, told that exploratory surgery have proven unsuccessful, and that their partner had very limited time to live:

> I remember crying a hell of a lot. But because it was just so quick… I was more shocked. I remember physically feeling sick to my stomach, and then it just sort of clicked… scared sick or something or something like that because I was literally physically sick. I just remember feeling really raw and empty, and just shocked. (4: bereaved)

> And I was with him then. His parents rang my mobile while I was with him… and they heard me tell him that they loved him… And then they hung up. (3: bereaved).

Children's worries about their parent's health manifested in several ways. One parent (5: survivor) had to sleep with their child to help keep them calm. One child resorted to prayer ('You know how children pray by the bedside… and she was going. Daddy, please don't die, please don't die' 6: survivor), while others acted out their stress at school ('A teacher contacted me and said, 'Your son's been in a bit of a mess in the class because he's worried about his mom'. 2: survivor).

For some family members, trauma continued in the form of flashbacks or memories of critical events for many months: 'I think now it's hitting me. Because I'm getting flashbacks and things like that. … And my mind's trying to deal with it. … And just sort a lying there, just shaking and not knowing who I am or where I am; and I'm not recognizing my room and not knowing my name.' (1: survivor).

#### Breaking the bad news

In the face of their own concerns, participants had to determine when and how to inform others of their partner's condition. Telling children was particularly challenging. Participants had to impart that their parent was sufficiently ill to require hospitalisation, but try to

maintain optimism and confidence (despite their own reservations) that they would return home:

> I was thinking, what do I say? Do I tell them? I thought it was probably best not to at this stage because they will not sleep… it'll just make them in a bit of a mess… I was like, 'Oh, well, she's staying in overnight, and everything will be all right, let's get to bed.' The other thing I thought about it was, what if I'm telling the kids, 'Everything's all right, everything's all right.', then in the morning, I'm going to get this message saying, your wife had a stroke or died in the night, and then I'm going to tell them this and actually she wasn't all right, and I was lying to them. (2: survivor)

### Waiting for news

Following admission to hospital, anxieties were often exacerbated by COVID-19-related visiting restrictions which resulted in very limited contact with clinicians. Relatives knew their condition was serious, but most struggled to track how their relative was progressing:

> … The biggest thing was not being with him. That was the hardest thing, and just waiting for… Because the hospital, they never rang me. The nurses, the doctors never rang me to say, 'This is what's happened.' (5: bereaved)

When achieved, communication could also be challenging and distressing for both participants and health professionals:

> … because they took all of the hope away from me. One particular doctor told me over the phone that all of the interventions they were doing, pretty much, were a waste of time and never called me back to talk me through that because I had to put the phone down. (8: survivor)

Communication with patients often involved the use of social media, which could be reassuring, but could also be stressful:

> I just laid here just waiting for the phone to go. Just waiting for a message saying, 'I'm fine', you know. He was losing all track of time and he'd send me messages at like 4:00 o'clock in the morning… It was just horrendous. All the things that go through your head. Just staring at the four walls and. Yeah, just waiting. (3: survivor)

The lack of direct contact with clinicians and patients led to anxieties about the veracity of patient reports about their health:

> I think there was always a worry that my mum was not going to be entirely 100% honest with it, not out of anything other than just like trying not to worry me. I found it hard… (7: survivor)

Finally, participants worried about the psychological impact of hospitalisation on the patient as well as experiencing their own stress:

> Whenever she's been ill or anything like that, I've always been with her. Not to be with her is quite upsetting because I'm then thinking of how scared and upset she's going to be… I came home and literally stood in the garden and cried. That's being military, 23 years in the army, quite hands-on, quite a person to get things done and fix things and make things right. It's horrible not being in control. (9: survivor)

### Theme 2: living with VITT
### Coming home is not the end

For survivors, initial relief following discharge was frequently replaced by the significant challenges of living with, and caring for, a very sick individual, who may have changed significantly from the person before admission to hospital.

> And the mood swings. You know, I think it's 'cos it's coming down off all the steroids and everything else. The mood swings that he had was… awful … just horrendous. He wasn't aggressive, but he just didn't realise how snappy he was getting…You just, you didn't know whether to speak, not speak. Do you go in? Do you not? You know, it was horrible. I remember my girl crying herself to sleep in my arms because it wasn't her dad that came home. (3: survivor)

Participants were concerned about the risk of future problems, and how to avoid them. They expressed concerns about their partner's health as well as awareness that medical aid was not immediately available:

> Yeah. I suppose when she came home, I was treating her with kid gloves. I felt like she was a bit, she was super-delicate, and I kept thinking don't bang your head, don't do anything when you knock your head or… And that probably got on her nerves a bit… (2: survivor)

For some, recovery from VITT meant a slow acceptance of improving health, confidence that they will not experience further health crises, and shifts towards a more normal life. For those nine individuals with continuing symptoms, the stresses and strains continued.

> I lie there at night listening to his breathing 'cos … he struggles a lot in the night. And some nights I nudge him to make sure he's going to take the next breath, cause he's really struggling to breathe, so I know if in the night his breathing is real bad I know in the morning, what kind of day it's gonna be. (3: survivor)

As well as the daily demands of caring, participants noted the toll of repeated hospital appointments (one participant noted their partner's 250th appointment had recently passed) either for monitoring, rehabilitation

or continuing investigations. Others family members reported the sheer stress of caring:

> … in the early days, it was a massively dark time in the fact that she was so angry and upset. She didn't want to be here anymore. Trying to keep her going, trying to keep her functioning, trying to get the right pills in her, trying to work out what's going on and trying to work out what appointment's coming next. … But we still have other days where it's dark and she gets angry and upset and she doesn't want to be here anymore. So, dealing with that is hard work… (9: survivor)

> I went through a really tired stage where I had this like chronic fatigue and I just, and it would just hit me wherever I was, and I'd actually have to go and lie down no matter what time of day it was. So that's exhaustion and fatigue. (1: survivor)

Most families reported little or no systematic support following discharge (although some individual clinicians were highly supportive) even when the patient had experienced significant life-changing conditions:

> …but there seems to be no sort of holistic approach to supporting families when a loved one has been through a life-changing condition in the hospital or in a coma. There's no pathway for that to support us. They just look at the patient… (8: survivor)

### The whole family has to adjust

The impact of the illness frequently affected the whole family. For some, as noted above, this involved daily commitments to care on a 24 hours a day basis. For others, the commitment may have been less, but life plans were nevertheless changed.

> My husband's job is in London, and we were thinking of moving up towards London. And after this happened, I decided let's buy a house here in X so I'm just three minutes away from my mum. And then my brother bought his and he's two minutes away from me. So, we're all literally in a line. So, we're all around to be here for her. (11: survivor)

At times, more subtle issues affected the whole family. Two participants, for example, reported the need to adjust the family day to day noise, and for their partner to have periods of quiet:

> … we can't behave the way we used to. Like before, we'd carry on as a normal day to day family messing around, noise, you know, you'd have all that … And now it's like you're kind of walking on egg shells all the time because you don't know… (6: survivor)

Another reported the adjustments necessary due to their partner's long-term fatigue:

> Now, we have to forward plan how busy they will be and his fatigue and all these things you haven't thought about before. Like swimming. We went swimming once and he was just walking in the water but only ten minutes. He had to get out with me, with the two children, and then he had to sleep. (8: survivor)

### Theme 3: multiple losses and some gains

Participants grieved multiple losses, and sometimes cried alone or together ('I try my best not to cry in front of my daughter… Other times I will cry to let her know that it is fine to cry. It's fine to let it out'. 3: survivor). For those whose partner died, these feelings were acute close to the event: 'The day after … I had the most excruciating stomach pain myself. And I hoped it was AstraZeneca-related and that it would finish me off too: I felt incredibly lonely…' (3: bereaved).

### Loss of a loved one

For those who were bereaved, the devastation was evident:

> It just seems so cruel. It's the cruelty of it. And it's the fact that it's irreversible. There's nothing. It's a help-lessness. That is the worst—there's nothing you can do to change this situation. (5: bereaved)

> I was on a lot of medication immediately following. I started having panic attacks, which I'd never had before. I was on antidepressants, sleeping tablets and diazepam. I'd never taken anything before. I'd always been really down to earth, grounded… I couldn't remember anything. I don't know if it's a combination of the drugs or the trauma or what, but I just couldn't remember a thing. I was sleeping a maximum of two hours a night, which was not helping anything. I didn't dream much, thank God, but when I did, it wasn't nice. … Yeah, just lots of very irrational ways. I couldn't read… That was really torturous. I can't do anything. I can't think. I can't watch TV. I can't take anything in. I can't hold a conversation. (2: bereaved)

Family members of survivors mourned the loss of the person they used to know ('It's almost a grief that you deal with' 9: survivor). For those with continuing health problems, outcomes such as fatigue, difficulties in mobilisation and low mood reduced not only the quality of life of the individual affected but also the quality of relationships with those around them.

> He's not the husband that I had and he's not the dad that he's always been and wants to be, and he's not the granddad that he's always thought he would be. … I mean, it's been what, two years… But since he's been out of hospital, there's just been a slow decline… And it's horrible to watch him just slipping away bit by bit… (3: survivor)

> Even now, the fatigue… She does an hour's work, and she's knackered. You don't see her before and after picture. I see the woman before that was sort of 8 stone, 9 stone, two kids, went out to work. Because I was out of work, she chose what she wanted to do so she'd do it. (5: survivor)

'Good days' are rare, but welcomed ('…you know, she knows like when he's on a good day, you know. She goes in and starts chatting away like she does. She always goes to him for the chats.' (5: survivor)), while roles within the relationship change and may even be reversed.

## Loss of a future

Another of the multiple losses these individuals experienced was the loss of an expected future:

> There's been some very dark, dark days and nights as such, where I really needed him to be the X that I know—and he can't so yeah, I just carry on… You just get up, you get dressed, and you see where the day goes and do what you can. And then you start all over again the next day. … I've never been wanting these massive holidays and do this do that, the nights out, you know. We've never really done all that, but just be able to just go out maybe for our anniversary … that would have been nice. (3: survivor)

One participant whose partner died summed up the experience of many, including those whose partner survived with continuing symptoms:

> We're thinking about slowing down. Enjoying your life now. You're getting older. In fact, X, he said, 'After lockdown ends, we'll go on holiday. We'll do something. Children are grown up now. It's our time.' So, it's all those things that are left with you, all those unfulfilled dreams, all those plans we had for the future. (5: bereaved)

Another stated:

> I'm still trying to do the things that give pleasure. But I realised the thing that gave me the most pleasure in my life was my marriage to my husband. And it feels like my world is never going to be the same because I don't have anyone in it who loves me like he did; nor do I have anyone to love in the way I loved him. (3: bereaved)

More pragmatic losses of future financial security were also evident, with many relying on redundancy, universal credit or Personal Independence Payment (PIP) to support them in constrained financial circumstances. One family sold a business, at a considerably lower than optimal price. Others survived on redundancy money that was gradually reducing over time, with consequent increasing limitations of the quality of life and availability of valued experiences.

## From trauma, growth

By contrast, a number of participants, particularly among the survivors, reported personal growth both as individuals and as a family, as a consequence of their experiences:

> I never knew that I could be so patient and compassionate… (6: survivor)

And I think having a near death experience possibly might have made me more positive about things, appreciating life a bit more… (3: survivor)

> It slowed us down and made us sit and think, there's more to life than this. There's kids, there's family. I mean, the family… Some good comes out of some of it. (5: survivor)

## Theme 4: battling against the system

One of the starkest issues reported by families of patients who died were 'battles' with the hospitals that cared, from their perspectives less than optimally, for their partner and the coroners involved in the death certification. Two made formal complaints about the hospital care. Three reported significant dissatisfaction with the process in the coroner's court, the medical representatives involved, or even their own legal team who failed to challenge some of the 'questionable' medical decisions and hospital processes ('…they weren't challenging the doctor who did the same-day referral. They weren't challenging… So, for me, I just felt, (expletive deleted), you are an absolute waste of space' (3: bereaved).

Three participants challenged death certificates which did not confirm VITT as cause of death, despite this being stated in medical notes: vital information, partly because partners wished the truth to be spoken about why their partner died, and more pragmatically because without a diagnosis of VITT on the death certificate they were not eligible for compensation under the Vaccine Damage Compensation Scheme (VDPS). Despite access to significant legal support, and in one case lobbying by both their General Practitioner (GP) and specialist nurse, participants found these challenges stressful and exhausting, an unwanted battle at a time they were grieving. As one person stated:

> And by that point, when the coroner rang me to say he died of natural causes. After the anger has subsided, after a day or two, I rang the coroner's office. And I said, 'I still don't agree with this. I want another inquest. I want…' And the chap … he kind of laughed at me. He said, 'It can take you years and can cost you a lot of money. They won't agree to this.' And there was not much more I could say, but I thought, 'I'm not letting this go. That's for sure.' (5: bereaved).

Another participant (3: bereaved) also noted: '… I need to complain about the coroners because nothing will change and I'm not convinced even if I do complain, anything will change. But if I don't, then it definitely won't'.

Even when the role of the vaccine was acknowledged by the coroner, this decision was not always immediate; and was only accessible after a struggle:

> I felt like I had to harass them to get any information and that it was very limited… We were just left wondering for 10 months. (1: bereaved)

Accessing the VDPS was initially very difficult and involved significant and excessive bureaucracy:

([VDPS] said, oh we need your husband's medical records. I said, don't be ridiculous. Why would you need his medical records? You've got a death certificate that says Astra Zeneca killed him. What more do you want? That's conclusive proof. (3: bereaved)

And even death did not necessarily fulfil the required level of disability:

The initial contact with the [VDPS] was to explain the situation and ask whether I'd be eligible. And the reply was basically, even if you can establish causation, death does not necessarily satisfy the 60% disability criteria! (1: bereaved)

Working through the VDPS was complex, and often ended in surviving patients failing to meet the criteria for funding, despite them being unable to work in either their previous or new employment, or receiving funding after considerable delays. This placed many in financial jeopardy. In addition, as noted above, even when patients had died, accessing the funding proved difficult.

For a number of participants, the need to publicise the risks associated with the vaccine was driven by anger at the lack of information at the time of vaccination:

So, I guess for a while I was just very angry that … you know… he wasn't forced to take it but then the kind of like realization that there wasn't any … how can they not inform us properly, how can they not tell me? Like, you know I'm not medical, so when they're telling me 'Have a vaccine', why aren't they telling me that there could be side effects? So, I was very confused and very sort of angry about. (4: bereaved)

As a consequence, some became deeply politically and emotionally involved in attempts to publicise the potential side-effects of vaccination through Instagram and other social media: posts and groups which were frequently taken down as they were identified as antivax, which, collectively, they were not:

I was telling people to get vaccinated. I was just saying, 'But be aware. If you have any side effects, don't do what we did. Don't be told, 'Oh, it's a migraine, you'll be fine,' because it's not. It could be something much more dangerous. (2: bereaved)

Likewise, mainstream media reported little or anything of their experiences:

I remember talking to an ex-ITV reporter who had got me in contact with ITV and This Morning, the programme, the news programme, and they had invited me on to share my story. So that's okay, great you know. And then I got a phone call from that reporter that said due to Ofcom regulations we can't have any story that would negatively impact the uptake of the vaccine. (4: bereaved)

So, publicity was largely achieved through multiple spots on GBNews, although this was viewed with some cynicism:

… It's like even the media outlets that give you a platform. It's always for a benefit of theirs you know. It's not really to help. It's to be alternative it's to be 'look what we're doing we're talking to these people'. (4: bereaved)

As a consequence, many of the family members of both deceased and surviving patients with VITT (and other vaccine damaged individuals) have formed a number of action groups, with the largest being VIBUK, in an attempt to influence the government and to instigate changes to the system, with support from one lawyer working *pro bono* on their behalf. Meetings have been held with parliamentarians, with some successes, although this brings its frustrations:

… when I watch Sir Christopher Choke talking in the House of Parliament and there's two people sitting there and neither of them are listening. They're on their phone. So, I feel I feel very you know, like they just don't want… there's no respect at all. There's no like, you know, if anyone was listening to somebody who had lost someone in any way in any capacity, there would be a human element to respond with compassion. (4: bereaved)

## DISCUSSION

This is the first study to report the psychosocial impact of the VITT beyond the directly affected individual. As always with qualitative studies, the sample was small and potentially biased through its sampling method, which was, initially at least, based on the Thrombosis UK twitter feed and closed membership of its VITT patient group. That said, the number of bereaved participants accounted for approximately 10% of all VITT-bereaved UK families.

The study captured a range of negative impacts on family members from both VITT and the acute exacerbation of distress as a consequence of the wider COVID-19-related hospital restrictions. Many of the outcomes were inevitable, and, while painful for those involved, could not be addressed by the health and legal systems. Others could. These include developing appropriate systems for supporting families during the in-patient period when direct access to clinicians and/or patients is not possible, preparation for very sick and disabled patients returning home, and attention to longer-term support for those with high caring burdens. Consideration also needs to be given to how both the coroners' courts and VDPS respond to people with significant needs and under significant stress, both emotional and financial. Finally, and echoing the findings in the previous study, the government needs to reflect on how to support and give voice to individuals and families in similar contexts in the future, should they arise.

**Contributors** PB was responsible for interviews and was lead in design, analysis and authorship of the study. FC conducted interviews, co-analysis and contributed to authorship of the study. JW conducted interviews and contributed to authorship of the study. BJH contributed to the design and authorship of the study. SP contributed to the design and authorship of the study. PB acted as guarantor.

**Funding** The authors have not declared a specific grant for this research from any funding agency in the public, commercial or not-for-profit sectors.

**Competing interests** None declared.

**Patient and public involvement** Patients and/or the public were involved in the design, or conduct, or reporting, or dissemination plans of this research. Refer to the Methods section for further details.

**Patient consent for publication** Not applicable.

**Ethics approval** This study involves human participants and was approved by Swansea University Department of Psychology. Approval number 5513-4722. Participants gave informed consent to participate in the study before taking part.

**Provenance and peer review** Not commissioned; externally peer reviewed.

**Data availability statement** Data are available upon reasonable request. Raw data (anonymised transcripts) are available from the corresponding author (PB).

**ORCID iD**
Paul Bennett http://orcid.org/0000-0003-2252-6065

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
