## [Reviewer comments · BMJ Open]

ARTICLE DETAILS

TITLE (PROVISIONAL)	A qualitative study of the consequences of vaccine-induced immune thrombocytopenia and thrombosis: the experiences of family members
AUTHORS	Bennett, Paul; Celik, Filiz; Winstanley, Jenna; Hunt, Beverley; Pavord, Sue

VERSION 1 – REVIEW

REVIEWER	Greinacher, Andreas Greifswald University Hospital
REVIEW RETURNED	20-Oct-2023

GENERAL COMMENTS	this is an interesting study and definitely requires publication. I have a few comments, mostly minor. Page 3, line 44: man age = mean age general comment: sometimes it is not clear for the reader whether statements of the affected patient or family member are reported. One example is the last paragraph on page 5. It reads like these are feelings of an affected patient. Another example is the first statement on page 9. Other statements are not related to VITT, but to Covid 19 related lock down quarantine measures. One example is the first statement on page 7. I recommend to remove these parts of the manuscript. For certain this had been a traumatic experience for everyone not specifically for VITT. The manuscript is already rather lengthy. It would benefit from removing parts which are not VITT specific. In the discussion or the introduction it can still be mentioned that in addition to the VITT related problems patients also suffered from restrictions associated with the pandemic, lock down etc.
--

REVIEWER	Bagot, C Glasgow Royal Infirmary, Dept of Haematology
REVIEW RETURNED	24-Oct-2023

GENERAL COMMENTS	This is a small qualitative study of the effect of VITTS on family members of those affected by the condition. Much of what the authors present would be applicable to any family member who has a relative affected by a life threatening condition, often with long term sequelae or resulting in death. This however, remains informative for healthcare professionals and the healthcare system for maintaining high standards of communication and ensuring support for individuals and their family members on discharge from hospital following a life threatening condition.
--

	Unique to VITTS is the effect on family members of difficulties in obtaining recognition and compensation for what occurred which needs to be recognised and which this publication succeeds in highlighting.
--	---

VERSION 1 – AUTHOR RESPONSE

Reviewer - page 3, line 44: man age = mean age –

Response: we have corrected and actually replaced with median age not mean.

Reviewer: sometimes it is not clear for the reader whether statements of the affected patient or family member are reported. One example is the last paragraph on page 5. It reads like these are feelings of an affected patient. Another example is the first statement on page 9. Response. All are family quotes. To reinforce this we have now indicated this more clearly in the text leading up to the quotes.

Reviewer: Other statements are not related to VITT, but to Covid 19 related lock down quarantine measures. One example is the first statement on page 7. I recommend to remove these parts of the manuscript. For certain this had been a traumatic experience for everyone not specifically for VITT. The manuscript is already rather lengthy. It would benefit from removing parts which are not VITT specific. In the discussion or the introduction it can still be mentioned that in addition to the VITT related problems patients also suffered from restrictions associated with the pandemic, lock down etc. Response: We understand the perspective of the reviewer. However, we also believe that while the quotes concerning the trauma of the hospital care were not necessarily specific to VITT, they added to the psychological burden of the participants and the report provides one of the few sources where this is documented. We have therefore cut out and shortened some of the quotes/text but left some in the document. We hope this is an acceptable compromise. We have also strengthened the discussion of the general patient versus VITT patient experience to really address this issue.

Editor: Please ensure that you have fully discussed the methodological limitations of the study in the Discussion section of the main text.

response: We have done so by adding details of the constraints noted earlier in the paper... "As always with qualitative studies, the sample was small and potentially biased through its sampling method, which was, initially at least, based on the Thrombosis UK twitter feed and closed membership of its VITT patient group. That said, the number of bereaved participants accounted for approximately 10 per cent..."

Finally, we note that the paper is over the recommended word count. We were not asked to reduce the word length in the requested revisions, and feel that shortening may reduce the quality and depth of the reported data. We hope this remains your view.

We hope these changes are acceptable and look forward to hearing from you in the fulness of time.